# Protein Stability Regulation in Osteosarcoma: The Ubiquitin-like Modifications and Glycosylation as Mediators of Tumor Growth and as Targets for Therapy

**DOI:** 10.3390/cells13060537

**Published:** 2024-03-18

**Authors:** Jacopo Di Gregorio, Laura Di Giuseppe, Sara Terreri, Michela Rossi, Giulia Battafarano, Olivia Pagliarosi, Vincenzo Flati, Andrea Del Fattore

**Affiliations:** 1Department of Biotechnological and Applied Clinical Sciences, University of L’Aquila, 67100 L’Aquila, Italy; jacopodigregorio@gmail.com; 2Department of Clinical, Internal, Anaesthesiological and Cardiovascular Sciences, Sapienza University, 00185 Rome, Italy; laura.digiuseppe@uniroma1.it; 3Bone Physiopathology Research Unit, Translational Pediatrics and Clinical Genetics Research Division, Bambino Gesù Children’s Hospital, IRCCS, 00146 Rome, Italy; sara.terreri@opbg.net (S.T.); michela1.rossi@opbg.net (M.R.); giulia.battafarano@opbg.net (G.B.); olivia.pagliarosi@opbg.net (O.P.); andrea.delfattore@opbg.net (A.D.F.)

**Keywords:** cancer, osteosarcoma, post-translational modifications, ubiquitination, NEDDylation, SUMOylation, glycosylation

## Abstract

The identification of new therapeutic targets and the development of innovative therapeutic approaches are the most important challenges for osteosarcoma treatment. In fact, despite being relatively rare, recurrence and metastatic potential, particularly to the lungs, make osteosarcoma a deadly form of cancer. In fact, although current treatments, including surgery and chemotherapy, have improved survival rates, the disease’s recurrence and metastasis are still unresolved complications. Insights for analyzing the still unclear molecular mechanisms of osteosarcoma development, and for finding new therapeutic targets, may arise from the study of post-translational protein modifications. Indeed, they can influence and alter protein structure, stability and function, and cellular interactions. Among all the post-translational modifications, ubiquitin-like modifications (ubiquitination, deubiquitination, SUMOylation, and NEDDylation), as well as glycosylation, are the most important for regulating protein stability, which is frequently altered in cancers including osteosarcoma. This review summarizes the relevance of ubiquitin-like modifications and glycosylation in osteosarcoma progression, providing an overview of protein stability regulation, as well as highlighting the molecular mediators of these processes in the context of osteosarcoma and their possible targeting for much-needed novel therapy.

## 1. Introduction

Osteosarcoma is a very aggressive primary bone cancer, targeting especially the young and pediatric population. It may develop in every bone, but the most common sites are the distal femur and proximal tibia, areas characterized by an elevated rate of bone turnover. Osteosarcoma has a very high metastatic potential, with the lung being the primary site of metastasis [1].

Despite it being the most common primary bone tumor, osteosarcoma is a relatively rare cancer, with an incidence of 3.4 per million cases per year; its pediatric onset, combined with its extremely high rate of recurrence and metastasis, makes osteosarcoma the object of intense investigation aimed to identify mediators of progression, invasion, and metastasis and new potential therapeutic targets and strategies. In fact, despite local surgery and chemotherapy having improved the survival rate of osteosarcoma patients (up to 65%, from less than 20% in the 70s), the recurrence rate of the disease, as well as its metastatic potential, is very high [2,3].

When examining molecular mechanisms that could play a dual role in both osteosarcoma progression and as potential therapeutic targets, post-translational modifications (PTMs) of proteins emerge as promising candidates.

PTMs, defined as covalent alterations of proteins after their synthesis [4], can modify the structure of a protein thus influencing its stability, function, cellular localization, and interactions. As a result, PTMs mediate and regulate several cellular processes and signaling pathways by influencing the stability and activity of their different target proteins.

Alterations of the fine mechanisms of the different PTMs can be associated with cancer progression in all its stages and are often responsible for the acquisition of the different hallmarks of cancer. This makes PTMs a very hot topic in cancer research.

In this regard, two of the most common and studied PTMs are phosphorylation, pivotal for the activation of a plethora of signaling pathways, and methylation, which can indirectly regulate the activity of transcription factors [5,6].

Other important PTMs involved in cancer progression are ubiquitination and ubiquitin-like modifications such as SUMOylation and NEDDylation. These PTMs can mediate protein stability or degradation, as well as protein–protein interactions. In addition, PTM can occur by glycosylation.

In the context of osteosarcoma, alterations of PTMs are a possible explanation of its pathogenetic mechanisms. Although focusing only on some PTMs may represent a limitation, we believe that this work gives the reader a focused overview of the mechanisms of protein stability regulation. Thus, this review will summarize how modifications in the processes of ubiquitination, ubiquitin-like modifications, and glycosylation are relevant for osteosarcoma progression and how they could be exploited for therapy.

## 2. Ubiquitination

Ubiquitination is a PTM characterized by the attachment on the lysine (K) residues of the target proteins of a small protein called ubiquitin (76 aa in length). This is achieved through a three-step reaction catalyzed by three different classes of enzymes. Briefly, the E1 ubiquitin-activating enzyme enables ubiquitin to be transferred to the substrate in an ATP-dependent way, the E2 ubiquitin-conjugating enzyme binds the activated ubiquitin, and the E3 ubiquitin ligase enzyme transfers ubiquitin from the E2 to the specific substrate (the process is summarized in Figure 1) [7]. Once attached to the substrate, ubiquitin can be further ubiquitinated, forming a poly-ubiquitin chain. Depending on the specific K residue that is poly-ubiquitinated on the ubiquitin sequence, the targeted protein can undergo different cell fates, ranging from degradation by the proteasome to increased stability. The different subtypes of ubiquitination are named after the target lysine residue on the ubiquitin sequence (K6, K11, K27, K29, K33, K48, and K63) [8]. Among them, the most studied are K48, linked to proteasomal degradation, and K63, related to increased stabilization. The other subtypes can be linked, other than degradation and stabilization, to autophagy and DNA damage response (K6), to mitotic regulation and cell cycle (K11), to innate immunity and mitochondrial DNA repair (K27), and to signal transduction and neurodegenerative disorders (K29) [8].

Of all the enzymes responsible for the process of ubiquitination, the E3 ubiquitin ligases are the largest family of the ubiquitin system (approximately, 600 are present in the human genome) and are the most studied. In fact, since they recognize and bind the substrates of the reaction, E3 ligases are pivotal for the specificity of ubiquitination and are identified (and referred to) as the enzymes responsible for substrate ubiquitination and are the last step of the ubiquitination cascade [9].

E3 ligases are divided into different classes, according to the structure of the respective active domain: HECT (homologous to the E6AP carboxyl terminus), RING (Really Interesting New Gene), U-Box, and RBR (RING-IBR-RING, sharing with HECT and RING the mechanistic aspect of ubiquitin-binding) [9]. Ubiquitination is mainly known for regulating protein stability, and for mediating proteasomal degradation (this usually happens through K48 ubiquitination [9]) and it regulates the turnover of cellular proteins involved in cell cycle regulation, DNA repair, and apoptosis, in normal cells. In cancer cells, these mechanisms can be dysregulated, leading either to reduced degradation of oncogenic proteins, or enhanced degradation of tumor suppressors. These dysregulations often happen due to the altered expression of E3 ligases or of other members of their regulatory complexes [10].

### 2.1. p53 Ubiquitination

One of the most important targets of ubiquitination is the K48 of the tumor suppressor p53 by the ubiquitin ligase mouse double minute 2 homolog (MDM2). As seen in several cancers, MDM2 activity is associated with cancer progression and evasion from apoptosis [11].

Osteosarcoma is not an exception: MDM2 levels are increased in the low-grade, low-differentiated forms of the malignancy, and have been proposed to be used as a marker to identify the more aggressive subgroups [12]. This means that p53 ubiquitination and degradation may be a crucial point for osteosarcoma progression and a possible focal point for targeted therapies [13]. However, MDM2 is not the only E3 ligase able to ubiquitinate p53 [14]. In fact, other E3 ligases such as constitutive photomorphogenesis protein 1 (COP1) [15], the carboxyl terminus of Hsp70-interacting protein (CHIP) [16], RING1 [17], topoisomerase-1 (TOPORS) [18], and Tripartite Motif Protein 24 (TRIM24) [19] have been found to mediate the degradation of p53 via ubiquitination. All these E3 ligases are differentially expressed in osteosarcoma [14,15,16,17,18,19], and in some cases, they are overexpressed so they can potentially contribute to p53 degradation (as well as of its homolog p73) and to osteosarcoma progression. However, further research is needed to prove this correlation. Other E3 ligases, known to target p53 for ubiquitination/degradation, are downregulated in osteosarcoma, and so they act as tumor suppressors. This is the case of HECT, UBA, and WWE Domain Containing E3 Ubiquitin Protein Ligase 1 (HUWE1), which inhibit osteosarcoma proliferation by inhibiting the Wnt signaling and by preventing accumulation of c-Myc [20,21], and of F-Box And WD Repeat Domain Containing 7 Protein (FBXW7), that blocks the cell cycle G1/S transition and proliferation in osteosarcoma cells [22]. For this reason, it is crucial to have a deeper understanding of the several substrates that can be regulated by E3 ligases in osteosarcoma, before considering them from a therapeutic standpoint.

### 2.2. PI3K/AKT/mTOR Pathway and Ubiquitination

The mammalian target of rapamycin (mTOR) plays an important role in osteosarcoma progression, as its pathway, the phosphatidylinositol 3-kinase (PI3K)/protein kinase B (AKT)/mTOR pathway, regulates a plethora of cellular processes, from cell growth and proliferation to metabolism and survival [23]. Briefly, mTOR operates within two distinct multiprotein complexes, mTOR Complex 1 (mTORC1) and mTOR Complex 2 (mTORC2), each one organized with specific molecular constituents and substrates [24].

In osteosarcoma, the mTOR pathway has been extensively studied, and it has been associated with cell proliferation, inhibition of apoptosis, autophagy inactivation, and metastasis initiation [25]. The mTOR pathway can be either positively or negatively regulated by the activity of different E3 ligases. Indeed, mTOR under the form of mTORC1 and mTORC2 can be stabilized, thus increasing their signaling, by K63 ubiquitination of either mTOR or other components of mTORC1 and 2.

TNF Receptor-Associated Factor 6 (TRAF6, a ubiquitin ligase related to NF-κB and MAPK pathways activation) is one of the several E3 ligases that can mediate the K63 ubiquitination and stabilization of the target protein [26], allowing mTORC1 activation. In osteosarcoma, TRAF6 has been found to be upregulated when compared to normal bone tissue, and it has been related to increased cell proliferation, reduced apoptosis, and enhanced invasive capacity [27]. Thus, silencing TRAF6 in osteosarcoma, or inhibiting its expression by using its regulator miR-146b-5p, represents a potential therapeutic approach.

Another E3 ligase that enhances mTOR signaling is the enzyme cullin 4 (CUL4), a member of the cullin-RING ligases family (CRLs) [28]. This enzyme exerts its E3 ubiquitin ligase activity on the regulatory associated protein of mTOR (RAPTOR), one of the components of the mTORC1 complex. This reaction further stabilizes mTORC1 and enhances its signaling activity [29]. In osteosarcoma, CUL4 is upregulated and acts as an oncogene [30], resulting not only in increased mTORC1 activation but also in the downregulation of tumor suppressors such as the phosphatase and tensin homolog (PTEN) and p21 in a K48-dependent manner [30]. In this regard, CUL4 silencing has been related to reduced proliferation and cell growth [31], and in osteosarcoma, this has been achieved using the TSC01682 small molecule, that acts as an inhibitor [32]. This makes CUL4, and its activity as an E3 ubiquitin ligase, a potentially valid therapeutic target.

mTOR signaling can be increased by K48-ubiqutination and degradation of DEP-domain containing mTOR-interacting protein (DEPTOR); a negative regulator of mTOR, able to act on both mTORC1 and mTORC2 [33]. DEPTOR is ubiquitinated either by the activity of the E3 ligases CUL5 [34] or Ring Finger Protein 7 (RNF7) [35]. This results in the increase of mTOR levels and in the inactivation of the autophagic process [36,37]. Although studies on the possible role of these enzymes in osteosarcoma are lacking, their effects suggest a tumor suppressor role for DEPTOR in osteosarcoma, as its upregulation should result in reduced mTOR pathway activity. However, an in vitro study on osteosarcoma cell lines showed opposite results, associating DEPTOR inhibition with reduced invasiveness and proliferation [38]. Thus, further research on DEPTOR is necessary to clarify its behavior in osteosarcoma and to possibly relate it to a therapeutic strategy.

mTOR and its signaling can also be downregulated by the activity of other E3 ubiquitin ligases. This is the case of FBXW7 [39], or Ring Finger Protein 126 (RNF126) [40]. They both target mTOR for K48 ubiquitination and are recognized as tumor suppressors in colorectal and hepatic cancers. FBXW7 has been identified as a tumor suppressor also in osteosarcoma and its high expression has been related to better patient survival, while in osteosarcoma cell lines the overexpression of FBXW7 causes apoptosis induction and growth arrest [22]. This makes FBXW7 a promising therapeutic target. On the other hand, RNF126 function has only been studied in cancer cell lines, and a clear understanding of this E3 ligase activity in osteosarcoma in vivo is lacking.

Furthermore, other components of the PI3K/AKT/mTOR pathway can be regulated by ubiquitination. In fact, both PI3K and AKT can be ubiquitinated and targeted for degradation.

PI3K is indeed targeted by the activity of the E3 ligase Makorin Ring Finger Protein 2 (MKRN2). In lung cancer cells, this results in reduced cancer migration and invasion, as the absence of MKRN2 leads to increased levels of both PI3K and AKT. In patients, MKRN2 absence is associated with poor prognosis [41]. This ligase has never been studied in osteosarcoma, thus further research is necessary before it can be taken into consideration for therapy purposes.

AKT exists in three highly homologous isoforms that can be ubiquitinated by different E3 ligases. Depending on the type of ubiquitination, this may result in the activation of the pathway, via K63 ubiquitination, or its inhibition via K48 ubiquitination [42]. Among all the AKT-activating E3 ligases, two are overexpressed in osteosarcoma. They are the S-Phase Kinase Associated Protein 2 (SKP2), whose inhibition has been recently linked to apoptosis induction [43], and the TNF Receptor-Associated Factor 4 (TRAF4), also upregulated in aggressive osteosarcoma [44] and linked to cancer cells proliferation and invasion. Both ligases can be activated by the methyltransferase SET domain bifurcate 1 (SETDB1), which is overexpressed in osteosarcoma [45] and that enhances the AKT signaling. Thus, targeting SETDB1 in osteosarcoma would represent a potential therapeutic strategy, as it would downregulate both SKP2 and TRAF4 (and their ability to activate AKT). On the other hand, when looking at the several E3 ligases known to directly downregulate AKT, there is only a little data about their potential involvement in osteosarcoma [46]. Then, investigating the role of E3 ligases such as CHIP, Tripartite Motif Protein 13 (TRIM13), or Zinc and Ring Finger 1 (ZNFR1) in osteosarcoma could uncover new mechanisms of cancer growth that could be targeted for treatment purposes.

An important regulator of the PI3K/AKT/mTOR pathway is the well-known tumor suppressor PTEN: as in other cancers, also in osteosarcoma PTEN has been found suppressed [47], and its restoration has been proposed and studied for therapeutic purposes.

Despite the genetic mechanisms being responsible for PTEN inactivation in osteosarcoma, the protein can be downregulated by ubiquitination. Indeed, E3 ligases known to target PTEN for K48 ubiquitination and degradation are upregulated or have an oncogenic function in osteosarcoma. They are members of the Tripartite Motif Protein (TRIM) family of E3 ligases comprising: TRIM10, associated with drug resistance [48]; TRIM14, which induces the epithelial-to-mesenchymal transition (EMT) and promotes cell proliferation [49]; TRIM59, involved in the escape from apoptosis and migration [50]. All these E3s contribute to PTEN degradation [51,52,53], and so their targeting should be considered to increase PTEN levels in osteosarcoma for therapeutic purposes.

### 2.3. NRF2 Pathway and Ubiquitination

The Nuclear factor erythroid 2-like 2 (NRF2) pathway is associated with osteosarcoma progression. NRF2 regulates the expression of antioxidant response element (ARE)-containing genes, which encode proteins involved in cellular antioxidant and detoxification pathways. NRF2 is regulated by the Kelch-like ECH-associated protein 1 (KEAP1), and its regulatory complex, including Cullin 3 and Ring-box 1 (KEAPeap1-Cul3-Rbx1, also called CRL3). KEAP1 functions as a substrate adaptor for Cul3-based ubiquitin ligase and facilitates the ubiquitination of NRF2 under normal cellular conditions. KEAP1 promotes the degradation of NRF2 by facilitating its ubiquitination and subsequent proteasomal degradation. Under conditions of oxidative stress, NRF2 cannot be recognized by the CRL3 complex, thus it translocates into the nucleus, where it activates the ARE-containing genes [54]. Increased NRF2 expression in osteosarcoma has been associated with poor prognosis [55], but also mutations of both NRF2 and KEAP1 have been reported in osteosarcoma [56]. KEAP1 is mostly found inactivated, disrupting its ability to ubiquitinate and degrade NRF2, and a mediator of this disruption is DDRGK domain-containing protein 1 (DDRGK1) [57], which has been shown to contribute to chemoresistance in in vitro models as well as in patients.

Instead, KEAP1 activation, for instance by using exosomes, has been shown to induce cell death in osteosarcoma, increasing NRF2 ubiquitination and degradation [58].

However, NRF2 can be ubiquitinated for degradation also by other E3 ligases such as TRIM22, which accelerates the ubiquitination and proteasomal degradation of NRF2 and increases autophagy but independently of KEAP1 activity, as seen in in vitro experiments on osteosarcoma [59]. Indeed, in osteosarcoma TRIM22 induction and the consequent NRF2 ubiquitination have been found to reduce progression and metastasis, making TRIM22 a target and a viable option for a synergic therapeutic strategy when coupled with KEAP1 induction [59]

### 2.4. HIF Pathway and Ubiquitination

The stability of the hypoxia-inducible factor (HIF), a central mediator of the response to low oxygen levels, is regulated by ubiquitination similarly to NRF2. The HIF pathway is crucial for regulating various cellular processes, such as angiogenesis, metabolism, and cell survival. The HIF family comprises the *alpha* (HIF-α) and *beta* (HIF-β) members. Under normoxic conditions, HIF1-α is hydroxylated at specific proline residues by prolyl hydroxylase domain proteins (PHDs). Hydroxylated HIF1-α is then recognized by the Von Hippel–Lindau protein (VHL) and bound to the Cullin 2 RING E3 ligase complex (CRL2), where it is ubiquitinated and subsequently degraded by the proteasome [60].

HIF1-α is a poor prognostic marker in osteosarcoma, and its downregulation has been associated with increased cell death [61,62]. In this scenario, the mechanisms that bring to the loss of HIF1-α ubiquitination, and to its consequent accumulation, are yet to be fully investigated, but promoting the process may lead to the HIF transcription factor downregulation and possibly to new therapeutic approaches for osteosarcoma.

Indeed, HIF1-α ubiquitination can be increased also by other ligases, this is the case of FBXW7, which is able to contribute to HIF degradation both in vitro and in vivo [63,64,65]. These data strengthen the tumor suppressor role played by FBXW7 in osteosarcoma.

Another enzyme able to increase HIF1-α ubiquitination, despite not being a ubiquitin ligase itself, is the receptor for activated C-kinase 1 (RACK1). RACK1 activity leads to HIF1-α ubiquitination and degradation, independently of the canonical PHD-VHL pathway [66]. However, RACK1 overexpression in osteosarcoma has been associated with increased proliferation, with a mechanism inhibited by the activity of TRIM26, another E3 ligase [67]. Thus, further studies are needed to determine if the degradation of HIF1-α by RACK1 would be exploitable for osteosarcoma therapy.

On the other hand, the activity of E3 ligases can also result in HIF1-α increase. For instance, the Seven in absentia homolog 1 and 2 (Siah 1–2), can target PHD for K48-mediated ubiquitination and this leads to increased HIF transcriptional activity in normoxia [68,69]. Siah1 has been found to be upregulated in osteosarcoma, where it increases resistance to doxorubicin treatment [70]; hence, a therapy targeting the HIF transcription factor in osteosarcoma should consider the inhibition of Siah1, to prevent its accumulation.

### 2.5. Other E3 Ligases and Ubiquitination

Other E3 ligase activity has been linked to increased cell proliferation, apoptosis evasion, and metastasis and is found upregulated in several osteosarcoma models.

For instance, silencing of the E3 F-box protein 39 (FBXO39) in osteosarcoma cell lines has been related to increased apoptosis [71], although the molecular mechanism has not yet been clarified. Similar results have been obtained when targeting members of the TRIM family of E3 ligases. TRIM11 silencing leads to ERK activation and reduced cell growth in vitro [72], whereas targeting TRIM66, both in vivo and in vitro, causes Transforming growth factor-beta (TGF-β) pathway activation, cell cycle inhibition, and apoptosis induction [73]. TRIM46 negatively regulates PPARα and can induce apoptosis inhibition in vitro [74]. TRIM58 is inactivated in osteosarcoma and its re-expression induces apoptosis via pyruvate kinase muscle isozyme M2 (PKM2) inhibition [75]. Thus, the TRIM protein family members are interesting therapeutic targets for osteosarcoma treatment.

Another interesting protein family from this standpoint, albeit less studied in osteosarcoma, is the SMAD Ubiquitination Regulatory Factor (SMURF) family of E3 ligases [76]. Indeed, SMURF1 can ubiquitinate SMAD1 in osteosarcoma cell lines, resulting in increased cell differentiation [77], whereas SMURF2 can reduce tumorigenesis via TRIM28 ubiquitination and degradation [78].

Taken together, these findings suggest that different families of E3 ubiquitin ligases may be involved in osteosarcoma molecular biology, and so their targeting is conceivable for therapeutic approaches.

An alternative strategy for osteosarcoma treatment that revolves around ubiquitination would involve proteasome inhibition, which essentially blocks the degradation induced by some subtypes of ubiquitination such as K48. This approach could represent an addition to the current therapy, especially in the neoadjuvant setting: indeed, different proteasome inhibitors (PI) have been tested in in vitro models and in limited clinical trials; the results are encouraging, even if the efficacy of PI in solid tumors is still low [79]. Hence, additional studies on the use of PIs for osteosarcoma treatment are needed to find the optimal treatment strategy.

## 3. Deubiquitination

Deubiquitinating enzymes (DUBs) are a class of enzymes able to remove an attached ubiquitin residue from a substrate, essentially counteracting ubiquitination [80]. DUBs regulate the recycling of ubiquitin after proteasomal degradation and can edit the length of ubiquitin modification by trimming and reducing the polyubiquitin chain [81]. The process of deubiquitination is summarized in Figure 2.

Almost a hundred DUBs have been identified, and they are divided into seven subfamilies including ubiquitin-specific proteases (USP), ubiquitin C-terminal hydrolases (UCH), ovarian tumor proteases (OTU), Machado–Josephin domain proteases (MJD), zinc-finger containing ubiquitin peptidase (ZUP1), Motif Interacting with Ub-containing novel DUB family (MINDY), and Jab1/Mov34/MPN + protease (JAMM).

Similarly to the altered ubiquitination, derangements in the deubiquitination process can lead to increased degradation of tumor suppressor proteins or accumulation of oncoproteins [82].

In osteosarcoma, several DUBs have been investigated and are reported to be involved with its progression. For the deubiquitination process, we will review the same pathways analyzed for the ubiquitination process, to show how the two processes are coordinated for the regulation of the ubiquitination targets and emphasize the necessity of evaluating both processes when designing therapeutic strategies that target ubiquitination.

### 3.1. p53 and MDM2 Deubiquitination

Reversing p53 ubiquitination contributes to the protein’s stability, hence exerting an anti-tumoral effect. Although data about the role of the DUBs in p53 stabilization are lacking in osteosarcoma, the potential role of deubiquitinases, such as OTU Deubiquitinase (OTUD) 1, 3 or 5, in p53 deubiquitination could be investigated [83] in order to achieve reduced p53 degradation and consequent cancer cell death.

However, the activity of DUBs can also result in increased stability of MDM2, thus resulting in further reduced levels of p53 [74]. Such DUBs include the Ubiquitin-specific proteases (USPs). For example, USP7 is known to be upregulated in osteosarcoma, but its ability to increase MDM2 levels in that context has not been investigated [84]. MDM2 can also be deubiquitinated by USP2, and this may contribute to osteosarcoma tumorigenesis [85]. A similar effect is induced by USP48, an enzyme belonging to the same family, albeit with a mechanism independent of its activity as shown in an in vitro osteosarcoma model [86] as well as in mice [87]. Overall, inhibiting the activity of these USPs may be a potential strategy for osteosarcoma treatment, although the detailed molecular mechanisms of their oncogenic activity need to be investigated.

### 3.2. PI3K/AKT/mTOR and Deubiquitination

Deubiquitination of AKT, and its following stabilization, would result in increased AKT signaling and tumor progression [46]. Indeed, the DUB ubiquitin C-terminal hydrolase 1 (UCHL1) functions through this mechanism. Its levels are increased in several metastatic cancers and its overexpression relates to increased AKT activation and reduced ubiquitination [88]. This is true also in osteosarcoma where a study performed on a subset of patients, has associated UCHL1 with a worse prognosis and increased metastasis. Subsequent investigation on osteosarcoma cell lines has confirmed that UCHL1 is related to tumor progression, migration, and metastasis, as well as to increased AKT activation. Furthermore, its inhibition can reduce tumor growth. This makes UCHL1 an oncogene, and a viable therapeutic target in osteosarcoma [89,90].

A similar role is played by another DUB, the USP7. In addition, to increasing AKT stability, this enzyme contributes to osteosarcoma progression by activating the Wnt signaling and the subsequent EMT [84]. USP7 also may increase the metastatic potential of AKT by deubiquitinating its target HIF-1α [91].

Furthermore, tumor progression can be increased by deubiquitinating PI3K, which functions upstream of AKT signaling. It is one of the various functions of USP14, whose inhibition has been proposed as a therapeutic strategy for osteosarcoma [92]. Downstream of AKT, instead, the activity of mTOR in osteosarcoma can be increased via deubiquitination by either OTU domain-containing 7B (OTUDB7), which increases mTORC2 activity by inhibiting TRAF2 [93,94], or by Ubiquitin-specific peptidase 9X (USP9X), able to enhance the signaling of both mTORC1 and 2 [95]. Both DUBs are upregulated in osteosarcoma so they could be targeted for therapeutic purposes.

### 3.3. NRF2 Deubiquitination

Given the oncogenic role played by NRF2 in osteosarcoma, and its degradation by ubiquitination, here proposed as a possible osteosarcoma therapy, the activity of DUBs, able to remove the ubiquitin chains from NRF2, represents a serious obstacle for the development of a therapeutic approach.

One of these DUBs is USP11, which increases the protein levels of NRF2 and enhances cell proliferation [96]. This DUB is upregulated in osteosarcoma cell lines and may promote tumor growth and metastasis [97,98].

However, deubiquitination can also result in lower levels of NRF2. For example, USP15 is able to deubiquitinate KEAP1 (that can be ubiquitinated by the same E3 ligase complex that ubiquitinates NRF2), resulting in lower protein levels of NRF2 [99]. USP15 is mutated or inactivated in several cancers [100,101], but data about its role in osteosarcoma are lacking, thus further research on USP15’s role in osteosarcoma is required to evaluate its possible targeting for therapy.

### 3.4. HIF Pathway and Deubiquitination

HIF-1α stability can be enhanced by the activity of DUBs. Similarly to the case of NRF2, the deubiquitinases can prevent the degradation of HIF-1α so their activity must be taken into consideration when developing a possible therapy based on the increased HIF-1α ubiquitination.

For example, the ubiquitination of HIF-1α mediated by FBXW7, analyzed earlier in this review, may be inhibited by the action of the Ubiquitin Specific Peptidase 28 (USP28), which acts as a FBXW7 regulator [102]. The function of USP28 has been studied in several cancers [103], and its oncogenic activity has been found to be mediated by HIF-1α stabilization as well as by increasing c-myc or c-Jun stability. However, its role in osteosarcoma is yet to be investigated.

Moreover, HIF-1α can be regulated by USP7, which is upregulated in osteosarcoma and has been found related to the activation of EMT [84]. Although USP7 must be K63 ubiquitinated (thus, further activated) to mediate this specific deubiquitination, this results in increased HIF-1α stability and activation, and in increased EMT as seen both in vitro and in vivo [91]. This, together with its other activities, makes USP7 a key target for osteosarcoma treatment, and so it is a focal point of research.

Another USP that stabilizes HIF-1α is USP19, as shown in in vitro studies [104]. Although USP19 has been linked to increased EMT [105] in other cancer models [106,107], its activity in osteosarcoma remains to be studied.

An enzyme with a similar mechanism of action is USP8, and its activity results in both HIF-1α and HIF-1β stabilization [108]. This DUB is associated with increased EMT, cancer progression, and metastasis in lung and breast cancer through the activation of the MAPK/JNK signaling pathway [109,110]. These data make USP8 another interesting object of research in osteosarcoma.

### 3.5. Other Deubiquitinating Enzymes of the USP Family

Other DUBs of the USP family are involved in osteosarcoma progression by acting either as tumor suppressors or oncogenes. For instance, USP1 slows osteosarcoma proliferation by deubiquitinating the inhibitors of DNA binding proteins (IDs) ID1, ID2, and ID3, in order to maintain a differentiated state in low-aggressive cells [111]. In addition, USP1 has been found to be able to inhibit the Hippo pathway in osteosarcoma cells by acting on the stability of the transcriptional coactivator with PDZ-binding motif (TAZ) [112], further confirming its role as a tumor suppressor in osteosarcoma. On the contrary, high expression of USP6 or USP41 in osteosarcoma cell lines and patients has been found to be related to an increased metastatic potential, and lower survivability [113], but the specific mechanism is yet to be investigated.

All these data, point out the importance of targeting the USPs and the DUBs for the development of new osteosarcoma treatments. However, a better understanding of the specific molecular mechanism by which the deubiquitinating enzymes act in osteosarcoma is necessary to design a more appropriate therapeutic approach.

## 4. SUMOylation

SUMOylation is a post-translational modification involving the attachment of small ubiquitin-like modifier (SUMO) proteins to specific target proteins. Since it is one of the most important ubiquitin-like PTMs, its involvement in cellular life has been extensively studied. This PTM plays a role in various cellular processes, including gene expression, DNA repair, and subcellular protein localization [114]. As in the case of the other PTMs, SUMOylation is a reversible and dynamic process. SUMO proteins have been found in all eukaryotic cells and they share a similar 3D structure with ubiquitin-like proteins even if just 20% of their amino acid sequence is homologous [115]. SUMO proteins are bound to cellular proteins in a covalent manner to modify their functions by mono-, multi-, or poly- SUMOylation. In mammals, 5 subtypes of SUMO proteins are expressed. While SUMO 1-3 are ubiquitously expressed, SUMO 4-5 have a tissue-specific expression. In particular, SUMO 4 is mainly found in kidneys, spleen, and lymph nodes, and SUMO 5 is prevalent in blood and testis [116,117].

Similarly to ubiquitination, SUMOylation is a multi-step reaction: the SUMO molecule is activated using ATP by the SUMO E1 activating enzyme, then transferred to the only known SUMO E2, the Ubiquitin carrier protein 9 (Ubc9). Then, a SUMO E3 ligase binds both Ubc9 and the specific substrate of SUMOylation, covalently attaching the SUMO molecule to a lysine residue of the targeted protein [118]. SUMOylation can be reversed (de-SUMOylation), by the activity of the Sentrin/SUMO-specific proteases (SENP 1–3 and SENP 5-7). Due to their function, SENPs are pivotal regulators of SUMOylation [114]. The process of SUMOylation is shown in Figure 3.

Alterations in the SUMO enzymes, as well as in the SENPs, have been found to be involved in the progression of several cancers [119,120]. Indeed, SUMOylation is known to regulate the activity of several tumor suppressor proteins, it can influence the cell cycle by modulating the activity of key regulatory proteins, it can be involved with DNA repair pathways, and it can affect the stability of oncoproteins [121].

The specific relationship between SUMOylation and osteosarcoma is an area of ongoing research but there are some papers highlighting the potential role of SUMOylation in osteosarcoma, as reported in the following sections of this review.

### 4.1. Altered Pathways and Processes by SUMOylation in Osteosarcoma

Alterations in the SUMOylation mechanisms can arise from altered levels of the enzymes at any point of the cascade. The only SUMO E2 enzyme, Ubc9 [118], is upregulated in osteosarcoma cell lines, such as U-2OS, where it behaves as an oncogene. Consistently, Ubc9 silencing in U-2OS cells reduces their proliferation and migration and increases apoptosis. Moreover, Ubc9 silencing improves the sensitivity of osteosarcoma cells to HSV-TK/GCV treatment both in vitro and in vivo [122]. The same study highlighted that de-SUMOylation of connexin 43 (Cx43), a component of gap junctions, causes the increase of free Cx43 levels, which is important for the recovery of physiological cellular functions, like cell integrity maintenance and cell adhesion. De-SUMOylation of Cx43, then, could be a strategy to improve the sensitivity of osteosarcoma cells to chemotherapy [123].

Inhibition of Ubc9 and SUMOylation in osteosarcoma can also cause a reduction in migration rate, as seen in an in vitro study [124]. The molecular mechanism involves the modulation of Talin, a key component of focal adhesions (FAs), important for cell migration, protein–protein interactions, and cell signaling. Talin is regulated by SUMOylation in osteosarcoma and, in this state, can decrease the number of FAs, increasing cell proliferation, migration, and invasion. Inhibiting the process of SUMOylation by targeting Ubc9 increases the number and size of FAs and ultimately reduces the migration capacity of osteosarcoma cells [124].

SUMOylation can also influence EMT. As mentioned earlier in this review, enhanced EMT in osteosarcoma has been associated with increased invasion, tumor microenvironment activation, and metastasis [105]. EMT can be activated by the overexpression of EMT activating factors and pathways, and in osteosarcoma, this mechanism could be mediated by high levels of the Zinc finger E-box-binding homeobox 1 (ZEB1), associated with increased proliferation and invasion and with poor prognosis in patients [125]. ZEB1 is further regulated by SUMOylation leading either to further activation or protein degradation, depending on the lysine residue that is SUMOylated [126]. Despite that direct evidence of ZEB1 SUMOylation in osteosarcoma is still missing, these data suggest that targeting ZEB1 by reducing its SUMOylation could possibly reduce EMT and invasiveness. However, further research is needed to confirm this hypothesis.

### 4.2. SENPs Alterations in Osteosarcoma

SENP activity can revert the SUMOylation process. Changes in the expression of SENPs can be linked to cancer progression because it alters the SUMOylation levels and the function of target proteins, thus making the SENPs interesting targets for therapy [120]. Given the tumor suppressive activity of SUMO inhibition in osteosarcoma, it would be expected that SENP activation would be directly related to cancer regression. However, the available studies about the role of SENPs in osteosarcoma describe a role more complex than anticipated. Depending on the single enzyme considered, SENPs may have an oncogenic or tumor-suppressive role. For example, studies on SENP2 have shown that the enzyme is downregulated in osteosarcoma cells, as well as in primary cancer tissues, and the induction of SENP2 results in reduced proliferation, migration, and invasion in osteosarcoma. The molecular mechanism involves the degradation of SRY-box-9 (SOX9), associated with increased aggressiveness in osteosarcoma [127]. If SUMOylated, SOX9 cannot be ubiquitinated and degraded, making the activity of SENP2 necessary for SOX9 degradation and the consequent reduced osteosarcoma cell proliferation, thus characterizing this SENP as a tumor suppressor [128].

Studies on SENP1 have highlighted a more complex role in cancer. SENP1 has been identified in vitro as the mediator of a positive feedback loop with HIF-1α, and its overexpression has been related to increased EMT and cell viability, while SENP1 inhibition reduces the levels of HIF-1α and the expression of its target genes; taken together, these data suggest an oncogenic role for SENP1 [129]. Consistently, SENP1 at a higher level has been found to be a poor prognostic marker in osteosarcoma patients [130]. However, another study has associated SENP1 overexpression with increased sensitivity to chemotherapy in osteosarcoma, as well as to reduced stemness and HIF-1α down-regulation [131]. Therefore, further studies on SENP1 are necessary to clarify its role in osteosarcoma.

On the other hand, SENP5 has oncogenic activity in osteosarcoma, both in patients’ primary cancer cells and in cell lines, where this SUMO protease is overexpressed, resulting in resistance to both apoptosis and arrest of the cell cycle. In fact, in vitro inhibition of SENP5 results in cell cycle arrest, due to the inhibition of cyclin B1 expression and increased apoptosis [132]. The mechanism of action of SENP5 in apoptosis evasion, as shown by subsequent studies, involves the de-SUMOylation of H2AZ, a key mediator of DNA damage response [133]. Thus, consistently with its activity as an apoptosis inhibitor observed in osteosarcoma, SENP5 mediates resistance to therapy, and on these bases, the inhibition of SENP5 may represent a possible strategy for the disease treatment.

Taken together, these studies show that the specific role of SUMOylation in osteosarcoma is likely influenced by several factors that can work in the opposite way. Research in this area is ongoing and understanding the molecular contribution of the SUMOylation of key proteins and pathways to the transformation of normal bone cells into cancer cells may provide potential targets for new therapeutic strategies for osteosarcoma. In this regard, inhibiting SUMOylation could represent a valid therapeutic approach. Indeed, a possible way to achieve SUMOylation inhibition in osteosarcoma could arise from the use of the small molecule TAK-981 (also called Subasumstat), an inhibitor of the SUMO E1 activating enzyme [134]. As seen in other cancers such as leukemia and lymphomas, TAK-981 administration has resulted in activation of the immune response, possibly leading to cancer regression [135,136,137]. However, studies on the use of TAK-981 in osteosarcoma are yet to be performed.

## 5. NEDDylation

NEDDylation is a PTM characterized by the attachment of the ubiquitin-like protein neural precursor cell expressed developmentally downregulated 8 (NEDD8) to a lysine residue of the target proteins [138].

NEDDylation is executed through an enzymatic cascade similar to ubiquitination and SUMOylation. The mature NEDD8 is adenylated and activated by the NEDD activating enzyme (NAE), then it is transferred to the NEDD8 conjugating enzyme, and finally bound to the target protein by a NEDD8 ligase (Figure 4) [139]. To date, there are ten recognized NEDD8 ligases. All of them also have E3 ubiquitin ligase activity and belong to the RING class of E3s [140].

NEDDylation can alter the stability of proteins, as well as their subcellular localization, and the ability to assemble in protein complexes [141].

### 5.1. Derangements in NEDD8 Ligases

Excessive NEDDylation can result in cancer progression and evasion from apoptosis [142]. This has also been observed in osteosarcoma, as NEDD8 ligases are upregulated in the disease. This is the case of MDM2 [12], which can NEDDylate p53, thus inhibiting its function without promoting its degradation [143]. Another NEDD8 ligases potentially involved in osteosarcoma progression is RNF111, which can activate the TGF-β pathway in order to increase EMT [144] or the inhibitors of apoptosis (IAPs). In osteosarcoma, the IAPs are upregulated and beyond their canonical role as inhibitors of the caspase cascade, can exert their functions by mediating NEDDylation [145].

### 5.2. Substrates and Inhibitors of NEDDylation

The best-known substrates of NEDDylation are the Cullin-RING ubiquitin ligases (CRLs), one of the largest families of E3 ubiquitin ligases. Upon NEDDylation, they are activated and mediate the ubiquitination of several substrates including transcription factors, cell cycle regulators such as cyclin B1, or DNA repair proteins such as p53, usually through the formation of a protein complex. Overall, CRLs are responsible for the ubiquitination and degradation of roughly 20% of the whole proteome [28]. In osteosarcoma, several CRLs are upregulated and their activation by NEDDylation is necessary for their oncogenic or tumor-suppressive activity. For instance, the CUL4 analogue CUL4B is highly expressed in osteosarcoma cells, and its activity, upon NEDDylation by RING-Box 1 Protein (RBX1), allows for degradation of p21, causing cell cycle progression [30,146]. This suggests an oncogenic role for the NEDDylated CUL4B in osteosarcoma.

Another CRL potentially interesting in the context of osteosarcoma is CUL7, which upon NEDDylation, is known to play an oncogenic role in other cancers, due to the upregulation of the EMT, as well as the induction of cell growth and invasion [147,148,149]. Direct evidence about the role of CUL7 in osteosarcoma is still missing, so further research on CUL7 is needed to uncover a new potential therapeutic target.

Besides cullin proteins, other important targets for NEDDylation, in the context of osteosarcoma, are p53 (NEDDylated by MDM2) [150], PTEN, which if NEDDylated can contribute to tumorigenesis [151], and SMURF1-2 that upon NEDDylation, are able to increase their E3 ligase activity that, as reported earlier in this review, inhibits osteosarcoma cell growth [152].

Despite the limited evidence about the activity of the NEDDylation enzymes in osteosarcoma, there are studies that suggest an important role for this PTM in cancer. In fact, MLN4924 (also known as pevonedistat), a molecule that inactivates the first step of the NEDDylation process and leads to complete inhibition of the PTM, has been proposed as a therapy for other cancers [153,154,155,156] and has been shown to be effective in osteosarcoma. In fact, treatment with MLN4924 results in cell cycle arrest, induced senescence, and increased apoptosis in osteosarcoma cell lines and tumor xenografts [157]. These effects are dependent on the inhibition of the cullin protein NEDDylation, and the accumulation of tumor suppressor proteins, highlighting the importance of this PTM for osteosarcoma progression. The same study has also shown a hyperactivation of the NEDDylation-activating enzyme NAE, and treatment with MLN4924 has shown similar results to the genetic inactivation of NAE. Further studies are necessary to confirm these data and to clarify the exact molecular mechanisms involved, but there are the foundations for targeting NEDDylation for osteosarcoma treatment.

## 6. Glycosylation

N-glycosylation and O-glycosylation are PTMs that involve the sequential addition of glycans to asparagine (Asn) and serine/threonine (Ser/Thr) residues, respectively, of target proteins [158]. N-glycosylation forms covalent linkages with branched sugars, while O-glycosylation initiates with the addition of monosaccharides like galactose, mannose, fucose, and N-acetylgalactosamine (GalNAc). Sialic acids typically cap both N- and O-linked glycans (Figure 5) [159,160].

Cell surface and extracellular molecule glycosylation play a crucial role in numerous molecular processes, including intracellular trafficking, protein quality control, cell interactions, and signal transduction [159]. Changes in glycosylation patterns contribute to diseases by incomplete synthesis and enhanced expressions of complex N-glycans, truncated O-glycans, overexpression of ‘core’ fucosylation and altered sialylation [161]. These alterations can facilitate the malignant transformation of cells. Consulting the Cancer Genome Atlas, gene expression alterations related to glycosylation in correlation with cancer progression can be identified; however, the underlying molecular mechanisms remain largely unexplored [162]. Recent studies indicate that changes in glycan structures are linked to stemness in cancer cells and EMT, as analyzed below. In the context of osteosarcoma, alterations of glycosylation are linked to progression and invasion.

### 6.1. Glycosylation and Cell–Cell/Cell–Matrix Interaction

The communication between epithelial cells relies on well-organized connections through tight and adherens junctions. E-cadherin, a calcium-dependent transmembrane protein, plays a crucial role in regulating cell adhesion, motility, and growth differentiation by forming a cadherin-catenin complex [163,164,165]. Disturbances in this complex, influenced by glycosylation changes, impact cell–cell interactions and cellular integrity in cancer cells. Examples include the upregulation of GnT-V affecting N-cadherin in fibrosarcoma cells and the destabilization of adherens junctions in oral cancer due to hyper-glycosylation of E-cadherin [166].

In osteosarcoma, there is limited evidence, but a study has correlated the altered expression of N-cadherin, due to a modulation of the sialyltransferase ST6Gal I activity to increased motility and invasiveness of osteosarcoma cells. Silencing the sialyltransferase decreases the expression of N-cadherin and metalloproteinases 2 and 9, as well as other EMT markers affecting osteosarcoma cells’ aggressiveness [167].

Alterations of glycosylation are extended to other proteins involved in cell–cell and cell–matrix interactions such as β-catenin, Intercellular Adhesion Molecule 1 (ICAM-1), Activated Leukocyte Cell Adhesion Molecule (ALCAM), and Mucin 1 (MUC1), impacting various cancer types and influencing invasion, metastasis, and cytoskeletal dynamics [168,169,170].

For example, the increased expression of N-acetylgalactosaminyltransferase 3 (GALNT3) leads to increased O-glycosylation of MUC1. This results in reduced stabilization of the E-cadherin/β-catenin complex, promoting cell proliferation and migration in ovarian cancer cells [170]. Conversely, inhibiting GALNT3 destabilizes MUC1, and stabilizes the E-cadherin/β-catenin complex, leading to the suppression of cell proliferation and invasion [171].

Recently, it was demonstrated that in osteosarcoma, glycosylation of molecules involved in the cell–matrix interaction plays a central role in tumor progression. Indeed, N-glycosylation of Procollagen C-proteinase enhancer protein (PCOLCE) is crucial for the development of lung metastasis in osteosarcoma. This PTM leads to increased protein stability and secretion into the ECM. RNA-seq analysis has shown the overexpression of PCOLCE in osteosarcoma tissues. Furthermore, PCOLCE enhances the activity of Bone Morphogenetic Protein 1 (BMP-1), a zinc metalloproteinase that modulates collagen deposition in the extracellular matrix. This is relevant because abnormal collagen deposition is a common ECM alteration during cancer progression, leading to increased cancer growth and metastasis [172]. Osteosarcoma cells transiently transfected with a PCOLCE mutant in the N-glycosylation site show a reduction in cell migration and metastasis, both in vitro and in vivo. This finding suggests that the glycosylation of PCOLCE is necessary for the promotion of osteosarcoma metastasis, and that targeting PCOLCE or its regulator Twist Family BHLH Transcription Factor 1 (Twist1) could be beneficial for metastatic osteosarcoma therapy [173].

### 6.2. Glycosylation and Oncogenic Signalling in Cancer

Altered glycosylation leads to modifications in cellular signaling and metabolism that can affect tumor progression. For example, altered glycosylation patterns on cell surface molecules and growth factors enhance tumor cell proliferation, invasion, and metastasis, through the activation of signaling cascades [174]. The oncogenic signaling pathways that could be involved in these processes are many, including Wnt/β-catenin, TGF-β/Smad, TNF/NF-κB, PI3K/AKT/mTOR, Hippo signaling, Notch signaling, Janus kinase/signal transducer, and activator of transcription (JAK/STAT) [174].

As mentioned in the previous paragraph, altered glycosylation affects the adherent junction complex with the simultaneous dissociation of tight junction proteins. These alterations affect the invasion, mediated by cell-matrix interaction, and promote metastasis of cancer cells by regulating the integrin and Wnt/β-catenin signaling. A recent study demonstrated that the N-glycosylation of a Wnt ligand, its receptors, and E-cadherin promotes the expression and nuclear translocation of β-catenin/γ-catenin that upregulates the transcriptional activity of dolichyl-phosphate N-acetylglucosamine phosphotransferase 1 (DPAGT1), which results in tumor progression and metastasis in oral cancer. Genetic deletion of DPAGT1 reduces the glycosylation of E-cadherin and downregulates the canonical Wnt signaling pathway, inhibiting tumor cell invasion and metastasis [163,164]. This mechanism could be exploited also in osteosarcoma, where the Wnt pathway is relevant for proliferation and invasion [175].

PI3Ks are crucial mediators of intracellular signaling in response to extracellular stimulators. Hyperactivation of PI3K signaling is among the most frequent events in human cancers. For instance, evidence suggests that N-glycosylation of β4-integrin contributes to cancer progression and cell migration by promoting PI3K/AKT signaling. Indeed, suppression of the N-glycosylation of β4-integrin by N-acetylglucosaminyltransferase III, GnT-III, is associated with reduced cell migration and tumorigenesis in MDA-MB435S cells expressing β4-integrin. Deletion of N-glycosylation sites in β4-integrin, which is accompanied by the downregulation of the PI3K signaling pathway, inhibits β4–integrin–dependent cell migration, invasion, proliferation, and tumor formation. Furthermore, the loss of association between galectin-3 and β4-integrin via β1,6GlcNAc-branched N-glycans abolishes galectin-3–promoting cancer cell adhesion to the extracellular matrix proteins and migration [176]. These data provide evidence that N-glycosylation of β4-integrin plays a functional role in promoting tumor development and progression through PI3K activation.

In osteosarcoma, different papers have reported the involvement of glycosylation in aberrant cellular signaling, leading to malignant transformation.

Alpha-(1,6)-fucosyltransferase (FUT8), is an enzyme responsible for core fucosylation. Aberrant fucosylation by the dysregulated expression of fucosyltransferases is responsible for the growth of various types of carcinomas [177,178,179,180]. FUT8 is expressed at reduced levels in osteosarcoma patients and in human osteosarcoma cell lines such as MNNG/HOS, U2OS, and 143B. This implies that a diminished expression of FUT8 plays a role in the growth and progression of osteosarcoma. In particular, FUT8 influences the survival of osteosarcoma cells by modifying the core-fucosylation of TNF receptors (TNFRs). The diminished fucosylation of TNFRs activates the non-canonical NF-κB signaling, subsequently reducing mitochondria-dependent apoptosis in osteosarcoma cells. Such findings indicate that FUT8 acts as a negative regulator of osteosarcoma cell survival, stimulating apoptosis [181]. Hence, it can be deduced that FUT8 plays a tumor suppressor role, and its induction may then represent a novel therapeutic strategy in osteosarcoma.

T-synthase core 1 beta 1,3-galactosyltransferase, (C1GALT1) is an enzyme involved in O-glycosylation, responsible for forming the core O-glycan. Its expression is upregulated in several cancers, but its role in osteosarcoma is still poorly defined. Up-regulation of C1GALT1 promotes the proliferation of osteosarcoma cells in vitro but exhibits a significant growth inhibitory effect after 3 weeks post-implantation in xenograft models *in vivo*. These effects could be explained by the fact that high C1GALT1 expression stimulated CD8+ T-cell proliferation and the increased production of IFN-γ by CD4+ T cell, inducing enhanced tumor lethality of Cytotoxic T lymphocytes (CTLs) and improving anti-tumor immunity [182]. Further studies on C1GALT1 activity in osteosarcoma are necessary to clarify its role.

As a matter of fact, glycosylation can be targeted for therapeutic purposes in osteosarcoma. For instance, treating the human osteosarcoma cell line Saos-2 with tunicamycin, a well-known N-glycosylation inhibitor, causes time-and dose-dependent decreases in cell-layer protein and cellular alkaline phosphatase (ALP) activity, demonstrating that the N-glycosylation process is essential for the synthesis and the distribution of ALP. This enzyme is essential for normal bone formation and mineralization. There is evidence that ALP expression in osteosarcoma cells could be inversely linked to their aggressiveness [183]. However, further studies are needed to elucidate if the loss of N-glycosylation could contribute to tumor progression in osteosarcoma. From a therapeutic standpoint, targeting the glycosylation process could achieve reduced cancer growth, also in osteosarcoma, even if specific inhibitors of this PTM are still under investigation [184].

## 7. Conclusions and Future Directions

The PTMs analyzed in this review have a strong impact on protein stability. Alterations in their mechanisms of action, in every step of their enzymatic cascades, can lead to increased cancer cell proliferation and cancer progression. In osteosarcoma, this is particularly relevant, as the need for additional therapeutic options, especially in advanced cancer, is high. Many potential lines of research could evolve into the development of therapies by either inhibiting a mediator of a PTM or restoring its activity. For example, FBXW7 (involved, as we have seen in this review, in mTOR and HIF ubiquitination) could be at the center of an activation strategy in osteosarcoma. This could be achieved by using a gene therapy approach or by developing a small molecule activating compound. Moreover, a combination with an inhibition strategy targeting USP28 (that de-ubiquitinates potential FBXW7 targets) could increase the effectiveness of FBXW7 activation. Similar strategies could be theorized for the several mediators that play or may potentially play, a role in osteosarcoma by acting on ubiquitination, SUMOylation, NEDDylation, or glycosylation.

Most of the PTMs analyzed in this review have shared targets and substrates, hence a combinatorial drug approach, that targets more than one PTM, would be also viable and potentially effective (an example in this regard is the E3 ligase CUL4, involved not only in the ubiquitination process but also in NEDDylation). However, further research on the post-translational modification processes is required to unravel their potential as targets for osteosarcoma treatment.

## Figures and Tables

**Figure 1 cells-13-00537-f001:**
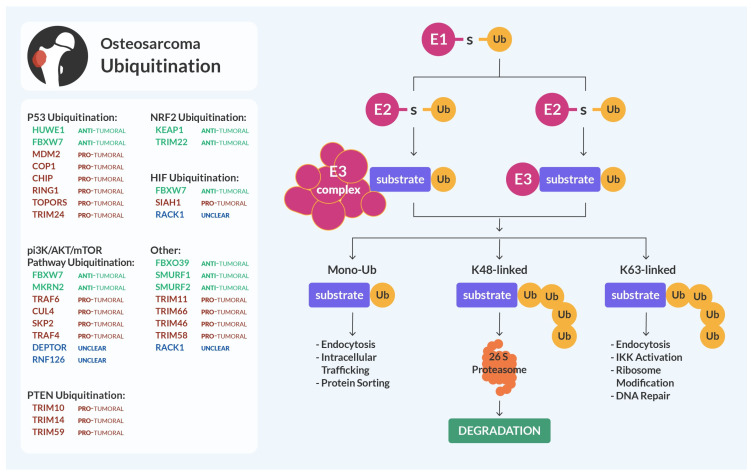
Ubiquitination and osteosarcoma. Ubiquitination of substrates of interest mediated by the indicated E3 ligases (either mono, K-48 linked, or K-63 linked as shown in the diagram) in osteosarcoma can be anti-tumoral (highlighted in green), pro-tumoral (in red), or unclear and needing further studies (in blue) to determine their precise role.

**Figure 2 cells-13-00537-f002:**
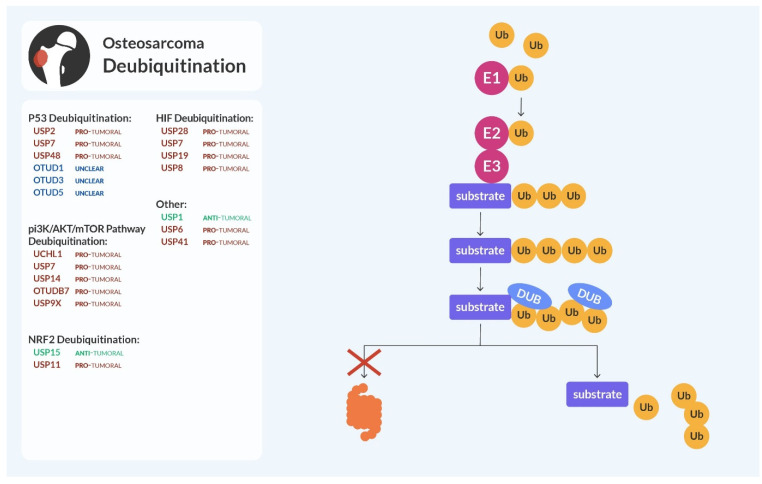
Deubiquitination and osteosarcoma. Removing the ubiquitin chains by the DUBs results in the inhibition of the proteasomal degradation of the ubiquitination substrates. The indicated DUBs in osteosarcoma, related to deubiquitination of substrates of interest, can be anti-tumoral (highlighted in green), pro-tumoral (in red), or unclear and needing further studies (in blue) to determine their precise role.

**Figure 3 cells-13-00537-f003:**
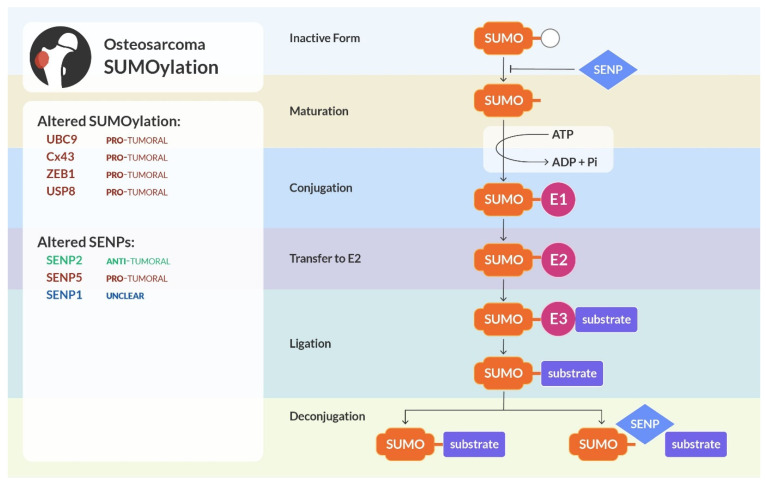
SUMOylation and osteosarcoma. Similarly to ubiquitination, SUMOylation is a cascade of enzymatic reactions, involving activation, conjugation, and ligation of the SUMO molecule. The action of SENPs removes SUMO from the substrate, allowing it to be attached to other proteins. Enzymes of the SUMO reaction and SENPs in osteosarcoma can be anti-tumoral (highlighted in green), pro-tumoral (in red), or unclear and need further studies (in blue) to determine their precise role.

**Figure 4 cells-13-00537-f004:**
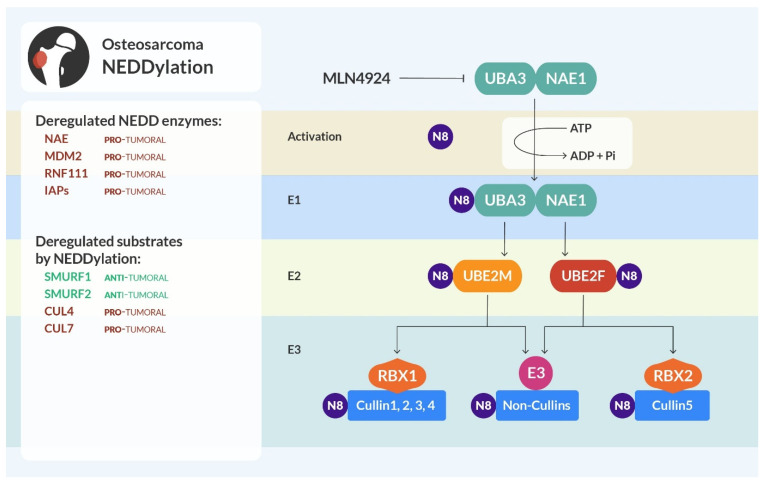
NEDDylation and osteosarcoma. NEDDylation is a cascade of enzymatic steps similar to ubiquitination: NEDD8 is activated, conjugated, and then attached to the substrate. RBX1 is the NEDD E3 ligase that mediates NEDDylation on Cul 1-4, and RBX2 NEDDylates Cul5. Enzymes of the NEDDylation cascade, as well as their potential substrates, in osteosarcoma can be anti-tumoral (highlighted in green) or pro-tumoral (in red).

**Figure 5 cells-13-00537-f005:**
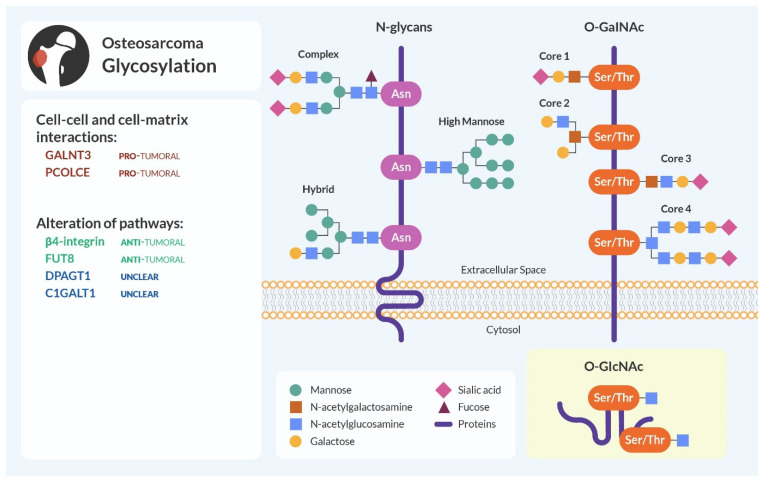
Glycosylation and osteosarcoma. N-glycosylation links branched sugars to Asn residues of the targeted protein, while O-glycosylation binds sugars to Ser/Thr residues. Glycosylation of specific proteins in osteosarcoma can be anti-tumoral (highlighted in green), pro-tumoral (in red), or unclear and needing further studies (in blue) to determine their precise role.

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
