# Peer review of "Protein Stability Regulation in Osteosarcoma: The Ubiquitin-like Modifications and Glycosylation as Mediators of Tumor Growth and as Targets for Therapy"

_cells, 2024, doi:10.3390/cells13060537_

Round 1

Reviewer 1 Report

Comments and Suggestions for Authors

The authors of this review aimed to summarize the role of protein stability in osteosarcoma progression, with a particular focus on ubiquitin-like modifications, including ubiquitination, deubiquitination, SUMOylation, NEDDylation, and glycosylation.

  1. 1 Given the concentration on osteosarcoma, signaling pathways unrelated to this cancer should be omitted from the discussion.

  2. 2 It is important to note that different subtypes of ubiquitination play distinct roles, which should be briefly introduced.

  3. 3 The title mentions "targets for Therapy," yet little information has been provided regarding therapeutic approaches.

Reviewer 2 Report

Comments and Suggestions for Authors

The authors review the literature on roles of post-translational modifications (PTMs) in osteosarcoma, with a focus on Ubiquitin family modifiers and glycosylation. Overall, this review provides a helpful compilation of information on the subject that will be of interest not only for biomedical researchers in the osteosarcoma field.

Being unable to evaluate the accuracy and completeness of all of the wide literature covered on the disease and its connections to the discussed PTMs, I focused mainly on the discussions of the individual PTMs and how they are controlled. Not in all exemplary cases, where I tried to find conclusions in papers referenced by the authors that would support their statements, I was able to find them (see below). The authors should therefore carefully double check whether their use of references is appropriate in all cases. 

Relating to chapter 4, in which involvement of sumoylation in the control of osteosarcoma and its treatment, it seems worthwhile to mention that recent advances leading to general inhibitors of the SUMO system (TAK-981/Subasumstat) have yielded promising results in the treatment of cancers. (Similar to what is mentioned in chapter 5 for the neddylation inhibitor MLN4924).

Below are some additional specific comments on details, some of which are semantic in nature:

Line 58: “..the structure of a protein thus influencing their stability..”: either “proteins” or “its stability”

Line 66-67: ..” and methylation, that can regulate the activity of transcription factors [5,6].” could be misunderstood as if the transcription factors are methylated.

Lines 69-69: It is probably oversimplification and not appropriate to say: “..modifications such as SUMOylation and NEDDylation. These PTMs usually mediate protein stability or degradation.” Probably more often, these PTMs regulate other processes by influencing protein interactions (discussed in more detail in chapter 4 of the manuscript).

Line 79: “..protein residue called ubiquitin..” The word “residue” does not fit here.

Line 88-89: “The different subtypes of [poly-]ubiquitination are named after the target lysine residue 88 on the ubiquitin sequence (K6, K11, K27, K29, K33, K48 and K63) [8]. It should be noted that not all of these chains lead to degradation, and that other non-lysine residues can be targets of ubiquitylation.

Line 90-91: “...the E3 ubiquitin ligases 90 are the most abundant..” could be understood as if these enzymes have the highest concentration, instead it is meant that they constitute the most diverse and largest family of enzymes in the system.

Line 98:”.. RBR (RING-IBR-RING, a hybrid between HECT and 98 RING) [9].”  Even though a similar phrasing is used in reference 9, it is not really appropriate. RBR ligase and HECT ligases only share the mechanistic aspect that they form a thioester intermediate with ubiquitin, but do not have anything else in common in sequence or structure.

Lines 152-155: “Another E3 ligase that enhances mTOR signalling is the enzyme cullin 4 (CUL4), a 152 member of the cullin-RING ligases family (CRLs) [28]. This enzyme exerts its activity on 153 the regulatory associated protein of mTOR (RAPTOR), one of the components of the 154 mTORC1 complex, also by K63 ubiquitination. This reaction further stabilizes mTORC1 155 and enhances its signalling activity [29]”. I could not find a mentioning of the K63 poly-ubiquitylation in the two references provided by the authors.

Lines 259-262: rephrase 2x “..bring to the…”

Lines 325-327:” For the deubiquitination process, we will review the same 325 pathways analysed in the ubiquitination section, to show how the two PTMs are con- 326 nected…”; It is confusing to suggest that ubiquitylation and deubiquitylation are two different PTMs. They rather together control the same PTM.

Lines 569-570:” Overall, CRLs are responsible for roughly 20% 569 the whole proteome ubiquitination and degradation [138].” Is this the correct reference?

Reference 128: It does not make sense to cite a Corrigendum, in which only the first author affiliation is corrected, rather than the original article.

Comments on the Quality of English Language

I a few places, some of which are mentioned in the details of my review, rephrasing is necessary and recommended.
